# Pathophysiological Response to SARS-CoV-2 Infection Detected by Infrared Spectroscopy Enables Rapid and Robust Saliva Screening for COVID-19

**DOI:** 10.3390/biomedicines10020351

**Published:** 2022-02-01

**Authors:** Seth T. Kazmer, Gunter Hartel, Harley Robinson, Renee S. Richards, Kexin Yan, Sebastiaan J. van Hal, Raymond Chan, Andrew Hind, David Bradley, Fabian Zieschang, Daniel J. Rawle, Thuy T. Le, David W. Reid, Andreas Suhrbier, Michelle M. Hill

**Affiliations:** 1Precision & Systems Biomedicine Laboratory, QIMR Berghofer Medical Research Institute, Herston, QLD 4006, Australia; s.kazmer@uqconnect.edu.au (S.T.K.); harley.robinson@qimrberghofer.edu.au (H.R.); srric0@eq.edu.au (R.S.R.); 2Biostatistics Unit, QIMR Berghofer Medical Research Institute, Herston, QLD 4006, Australia; gunter.hartel@qimrberghofer.edu.au; 3Inflammation Biology Laboratory, QIMR Berghofer Medical Research Institute, Herston, QLD 4006, Australia; Kexin.Yan@qimrberghofer.edu.au (K.Y.); Daniel.Rawle@qimrberghofer.edu.au (D.J.R.); Thuy.Le@qimrberghofer.edu.au (T.T.L.); Andreas.Suhrbier@qimrberghofer.edu.au (A.S.); 4New South Wales Health Pathology-Royal Prince Alfred Hospital, Camperdown, NSW 2050, Australia; sebastiaan.vanhal@health.nsw.gov.au (S.J.v.H.); raymond.chan1@health.nsw.gov.au (R.C.); 5Agilent Technologies Australia, Mulgrave, VIC 3170, Australia; andrew.hind@agilent.com (A.H.); david.bradley@agilent.com (D.B.); fabian.zieschang@agilent.com (F.Z.); 6Lung Inflammation & Infection Laboratory, QIMR Berghofer Medical Research Institute, Herston, QLD 4006, Australia; david.reid@qimrberghofer.edu.au; 7The Prince Charles Hospital, Chermside, QLD 4032, Australia; 8Australian Infectious Disease Research Centre, GVN Centre of Excellence, Brisbane, QLD 4029, Australia; 9UQ Centre for Clinical Research, Faculty of Medicine, The University of Queensland, Herston, QLD 4006, Australia

**Keywords:** fourier transform infrared spectroscopy, FTIR, COVID-19 pandemic, kallikrein

## Abstract

Fourier transform infrared (FTIR) spectroscopy provides a (bio)chemical snapshot of the sample, and was recently used in proof-of-concept cohort studies for COVID-19 saliva screening. However, the biological basis of the proposed technology has not been established. To investigate underlying pathophysiology, we conducted controlled infection experiments on Vero E6 cells in vitro and K18-hACE2 mice in vivo. Potentially infectious culture supernatant or mouse oral lavage samples were treated with ethanol or 75% (*v*/*v*) Trizol for attenuated total reflectance (ATR)-FTIR spectroscopy and proteomics, or RT-PCR, respectively. Controlled infection with UV-inactivated SARS-CoV-2 elicited strong biochemical changes in culture supernatant/oral lavage despite a lack of viral replication, determined by RT-PCR or a cell culture infectious dose 50% assay. Nevertheless, SARS-CoV-2 infection induced additional FTIR signals over UV-inactivated SARS-CoV-2 infection in both cell and mouse models, which correspond to aggregated proteins and RNA. Proteomics of mouse oral lavage revealed increased secretion of kallikreins and immune modulatory proteins. Next, we collected saliva from a cohort of human participants (*n* = 104) and developed a predictive model for COVID-19 using partial least squares discriminant analysis. While high sensitivity of 93.48% was achieved through leave-one-out cross-validation, COVID-19 patients testing negative on follow-up on the day of saliva sampling using RT-PCR was poorly predicted in this model. Importantly, COVID-19 vaccination did not lead to the misclassification of COVID-19 negatives. Finally, meta-analysis revealed that SARS-CoV-2 induced increases in the amide II band in all arms of this study and in recently published cohort studies, indicative of altered β-sheet structures in secreted proteins. In conclusion, this study reveals a consistent secretory pathophysiological response to SARS-CoV-2, as well as a simple, robust method for COVID-19 saliva screening using ATR-FTIR.

## 1. Introduction

Following the initial alarm of a cluster of atypical viral pneumonia in the city of Wuhan, China, on 31 December 2019, severe acute respiratory syndrome coronavirus 2 (SARS-CoV-2) rapidly spread around the world, leading the World Health Organization to declare coronavirus disease 2019 (COVID-19) a public health emergency of international concern (PHEIC) on 30 January 2020, and a global pandemic on 11 March 2020 [1,2]. As of 7 January 2022, close to 300 million confirmed COVID-19 cases and ~5.5 million COVID-19 deaths have been reported to WHO [3]. These numbers are likely under-reported, as developing countries/regions and rural populations may not have ready access to testing facilities or at-home tests.

While the current gold standard, reverse transcriptase quantitative polymerase chain reaction (RT-qPCR), is highly sensitive in detecting SARS-CoV-2 RNA, the technical requirements, time-to-result, and the accumulated testing costs are prohibitive in developing countries and disadvantaged communities. Globally, numerous technological innovations have been explored for rapid, portable, and economic COVID-19 testing to fulfil the continued and evolving testing needs as the global pandemic scales new waves. For example, several teams have used isothermal target nucleic acid amplification and CRISPR-Cas enzyme systems for rapid SARS-CoV-2 detection from RNA extracts (reviewed in [4,5]). Indeed, equipment-free rapid antigen tests with immobilized anti-SARS-CoV-2 antibodies in lateral flow devices are now widely used in some countries. These tests can be performed at home with self-collected samples and generate results in 5–20 min, with the caveats of potential sampling errors and unknown reactivity to new variants. Evaluations of currently available rapid antigen tests have reported variable sensitivities [6,7]. With the global research community actively investigating the use of novel nanomaterials and chemistries for rapid antigen tests, sensitivities and ease of use will likely improve in the near future.

As an alternative to using specific reagents (i.e., PCR primers, antibodies) to detect SARS-CoV-2 viral RNA or proteins, Fourier transform infrared (FTIR) spectroscopy was recently reported as a promising, point-of-care technology for COVID-19 detection using a pharyngeal swab or saliva [8,9,10]. FTIR provides a biochemical snapshot of the sample by measuring the vibration states of chemical bonds [11]. FTIR spectra collected from the saliva of COVID-19 patients and healthy controls have been used to develop prediction algorithms that demonstrate high predictive accuracy in the cross-validation of the same cohort [9,10] or in an independent cohort [8]. FTIR sampling, using either transflection (slide mount) or attenuated total reflectance (ATR, directly deposited onto highly reflective crystal), was able to distinguish healthy controls from confirmed COVID-19 cases with high specificity and sensitivity [8,9,10]. These recent cross-sectional cohort studies provide promising proof-of-concept for the use of FTIR in COVID-19 screening using saliva as a non-invasive sample. However, a biosafe sample processing method remains to be developed, no controlled infection experiments using FTIR have been reported, and there is a lack of knowledge on the biological basis of the proposed saliva diagnostic. The goal of this paper is to address these three research/knowledge gaps to enable the translation and use of FTIR technology in response to the ongoing global COVID-19 pandemic.

For biosafe ATR-FTIR spectroscopy, we recently reported a decontamination procedure with the addition of 100% ethanol to plasma to obtain 75% final (*v*/*v*), and used this method in developing ATR-FTIR for predicting COVID-19 severity using plasma samples [12]. The high ethanol percentage facilitated the rapid evaporation of the treated plasma (1 μL) on the ATR-FTIR target (~30 s), thus enabling very rapid data acquisition [12]. Here, we applied the ethanol decontamination procedure to experimental infection and participant saliva samples to investigate the pathophysiology underlying FTIR-based saliva screening. We utilized three different biological systems: culture supernatant from in vitro cell infection, oral lavage of inoculated hACE-2 mice, and human saliva from a cohort of COVID-19 patients and health controls. For the cell and mouse models, UV-inactivated SARS-CoV-2 virus was used as control, and two post-infection time points were examined. In addition to ATR-FTIR analysis of cell culture supernatant, mouse oral lavage, and human saliva samples, we also conducted mass spectrometry-based proteomics on the mouse oral lavage samples, in order to decipher the underlying pathology and to correlate with the FTIR spectra. Finally, we used the human cohort data to develop a COVID-19 saliva FTIR signature, and conducted a meta-analysis of all available COVID-19 FTIR spectra data to determine the robustness of the response.

This systematic investigation contributes several important findings to the field. Firstly, the controlled cell and mouse model infection experiments provided proof-of-specificity for a SARS-CoV-2 infection ATR-FTIR signature in secretions. The novel oral lavage proteomics data from the control- and SARS-CoV-2 infected mice provides novel insight into the physiological changes detectable in the oral fluid during SARS-CoV-2 infection, which also enabled biological interpretation of the COVID-19 saliva signature. In agreement with previous studies, the predictive model developed from our human cohort data showed high sensitivity and specificity for detecting COVID positivity, confirming the validity of our biosafe procedure and the FTIR diagnostic technology. However, our study revealed that COVID patients testing negative on the day of saliva collection (i.e., recovered patients) could not be easily distinguished from COVID-19 positive patients based on saliva FTIR, which has implications for the intended use of saliva FTIR for population screening. Finally, our meta-analysis of all available saliva/secretion COVID FTIR signatures from our study and the literature demonstrated overlap in several regions, thereby confirming the robustness of the technology.

## 2. Materials and Methods

### 2.1. SARS-CoV-2 Virus

The SARS-CoV-2 isolate (hCoV-19/Australia/QLD02/2020) was provided by Dr Alyssa Pyke (Queensland Health Forensic & Scientific Services, Queensland Department of Health, Brisbane, Australia). Virus stocks were prepared in Vero E6 cells as previously described [13], with all infectious SARS-CoV-2 work conducted in a dedicated suite in a biosafety level 3 (PC3) facility at the QIMR Berghofer MRI (Australian Department of Agriculture, Water and the Environment certification Q2326 and Office of the Gene Technology Regulator certification 3445). An aliquot of the viral stock was extracted and sequenced using Illumina technology and uploaded to GISAID (https://www.gisaid.org, accessed on 6 February 2020) under Accession ID (EPL_ISL_407896).

### 2.2. In Vitro Cell Infection Model

Vero E6 (C1008, ECACC, Wiltshire, England; obtained from Sigma-Aldrich) were maintained in RPMI 1640 (Thermo Fisher Scientific, Waltham, MA, USA), supplemented with endotoxin-free, 10% heat-inactivated fetal bovine serum (FBS; Sigma-Aldrich, St. Louis, MO, USA) at 37 °C, and 5% CO_2_. Cells were checked for mycoplasma using the MycoAlert Mycoplasma Detection Kit (Lonza, Basel, Switzerland). FBS was checked for endotoxin contamination before purchase, as previously described [14].

Vero E6 cells (6 × 10^5^ total cells) were plated onto 6-well plates in 2 mL RPMI + 10% FBS without phenol red (to limit background spectra). Following 24 h of growth, media was removed and cells were subjected to control, mock-infection, or infection regimes, each in triplicate. Control (untreated) cells were rinsed with 2 × PBS and placed in 3 mL phenol red RPMI + 2% FBS. Mock-infected cells were incubated with 500 µL of UV-inactivated SARS-CoV-2 stock for 30 min, while infected cells were incubated with 500 µL SARS-CoV-2 viral stock MOI 0.01 for 30 min. Mock-infected and infected cells were then rinsed with 2 × PBS and placed in 3 mL phenol red RPMI + 2% FBS. Aliquots of conditioned media were collected at 24 and 48 h post-infection. At each time point, 100 µL of conditioned media was mixed with 300 µL ice cold 100% ethanol for FTIR, while 200 µL conditioned media was mixed with 600 µL Trizol-LS for PCR. At the 48 h time point, remaining supernatant was discarded and cells were harvested in 400 µL Trizol-LS for PCR.

### 2.3. Nucleic Acid Extraction and RT–qPCR

RNA was purified from tissue culture supernatants and saliva (TPCH and QIMRB cohorts) using the Direct-zol RNA MicroPrep Kit (Zymo Research, Irvine, CA, USA) and cDNA was generated using iScript™ Reverse Transcription Supermix (Bio-Rad, Hercules, CA, USA). For qPCR, SsoAdvanced™ Universal SYBR^®^ Green Supermix (Bio-Rad, Hercules, CA, USA) was used with two previously published primer sets targeting different regions of SARS-CoV-2: 1) forward (5′-CAATGCTGCAATCGTGCTAC-3′) and reverse (5′-GTTGCGACTACGTGATGAGG-3′) primers targeting the N-gene; 2) forward (5′-ACCTTCCCAGGTAACAAACCA-3′) and reverse (5′-TTACCTTTCGGTCACACCCG-3′) primers targeting the 5’UTR. Cycling was carried out in a CFX384 Touch™ Real-Time PCR Detection System (Bio-Rad) under the following conditions: 95 °C, 30 s; 95 °C, 10 s; 60 °C, 30 s (40×); melt curve 65–95 °C. Viral copy number in experimental samples was estimated relative to a reference cDNA standard, using primer set 1. The reference cDNA was generated from a pool of SARS-CoV-2 infected Vero E6 cell supernatant RNA, and the viral copy number of reference cDNA was estimated relative to a plasmid containing the 5’UTR of SARS-CoV-2 (provided by Dongsheng Li, QIMR Berghofer MRI), using primer set 2. Plasmid copy number was determined using the URI Genomics and Sequencing Centre online calculator (http://cels.uri.edu/gsc/cndna.html, accessed on 29 December 2021). Saliva RNA quality was confirmed by amplification of housekeeping gene, β2-microglobulin, using forward (5′-ACTCTCTCTCTTTCTGGCCTGG-3′) and reverse (5′-CATTCTCTGCTGGATGACGTG-3′) primers.

### 2.4. Mouse Model

All mouse work was conducted in accordance with the Australian code for the care and use of animals for scientific purposes, as defined by the National Health and Medical Research Council of Australia. Mouse work was approved by the QIMR Berghofer Medical Research Institute animal ethics committee (P3600, A2003-607, project approved 8 April 2020). K18-hACE2+/− mice were purchased from Jackson laboratories and were maintained in-house as heterozygotes by backcrossing to C57BL6/J mice [13,15]. Mice were assorted as described in [13] using forward (5′-CTT GGT GAT ATG TGG GGT AGA -3′) and reverse (5′-CGC TTC ATC TCC CAC CAC TT -3′) hACE2 primers (recommended by NIOBIOHN, Osaka, Japan).

Prior to oral lavage, 4–5-month-old K18-hACE2+/− mice were placed into static microisolator cages (Techniplast Static Micro-isolator # 1264) with grid floors to allow feces to pass through, for 1 h without food or water, to avoid fecal contamination in the oral cavity. Oral lavage was conducted with the mice under light anaesthesia: 3% isoflurane (Piramal Enterprises Ltd., Andhra Pradesh, India) was delivered using The Stinger Rodent Anaesthesia System (Advanced Anaesthesia Specialists/Darvall, Gladesville, NSW, Australia). With the mouse lying on its back, 25 µL of milliQ water was placed into the side of the mouth just behind the teeth of the lower mandible. The water was pipetted up and down 4 times to wash the mouth without injury or abrasion of gums or lips. The lavage was recovered, and 15 µL was added to 45 µL of 100% ethanol to obtain 75% (*v*/*v*) ethanol, then stored at −80 °C.

Mice were infected via the intrapulmonary/nasal route with 5 × 10^4^ CCID_50_ SARS-CoV-2 in 50 μL medium while under light anaesthesia. Saliva samples were collected before infection and on the indicated days after infection. Mouse body weight was measured each day. Mice were euthanized using CO_2_ on day 4 post-infection, and lung titers determined by CCID_50_ assay performed on serial dilutions of supernatants from homogenized lung tissues [13].

### 2.5. Mouse Oral Lavage Proteomics

Protein precipitates were collected from ethanol-treated lavage samples by centrifugation at 16,000× *g* for 25 min at 4 °C. Supernatant was discarded, and the protein pellet washed twice with 50 mM triethylammonium bicarbonate (TEAB, Sigma-Aldrich) buffer. Proteins were resuspended in 50 mM TEAB and underwent protein estimation by BCA assay, as per manufacturer’s instruction (Thermo Fisher Scientific). Outliers were defined as samples with a protein abundance 3 standard deviations above the mean for that condition, and were excluded from further processing. The resulting day 0 lavage protein samples were pooled due to their low abundance, resulting in a single proteomics sample. Day 4 lavage samples contained adequate protein abundance to continue as individual replicates, *n* = 4 for SARS-CoV-2^UV-I^ and *n* = 8 for SARS-CoV-2^POS^ conditions. The BRAVO AssayMap platform (Agilent Technologies) was used for in-solution digest and C18 desalting procedures. 1% sodium deoxycholate was added to each sample for increased protein solubility. A standard automated trypsin digest method was followed using 5 mM dithiothreitol and 20 mM 2-iodoacetamide. Samples were diluted 1:10 with 50 mM TEAB and porcine trypsin (Promega) added (final 1:30 trypsin to sample protein ratio). Digests were incubated overnight at 37 °C and acidified using trifluoroacetic acid (TFA) to a final concentration of 0.5%. Sodium deoxycholate was pelleted by centrifugation for 30 min at 5000× *g* at room temperature. Peptides were subsequently desalted using AssayMAP C18 cartridges, following manufacturer’s instruction. Eluted peptides were dried and resuspended in 0.5% TFA.

Peptides were resolved on a Thermo U3000 nanoHPLC system and analyzed on a Thermo Q Exactive Plus Orbitrap mass spectrometer. The HPLC setup used a C18 trap column and a 50 cm EasySpray C-18 analytical column (Thermo Fisher, catalogue: 160454, ES803A). Mobile phases were A (0.1% formic acid) and B (80% acetonitrile with 0.1% formic acid). The loading pump ran on 3% B at 10 μL per minute. Subsequently, 1 µg peptide was loaded in 3% B. The nano-capillary pump ran at 250 nL per minute, starting at 3% B. The multi-step gradient was as follows: 3% to 6% B over 1 min, 6% to 30% B over the following 60 min, 30% to 50% B over the following 12 min, then 50% to 95% B over 1 min. After maintaining 95% B for 12 min, the system was re-equilibrated to 3% B. The mass spectrometer ran an EasySpray source in positive ion DDA mode, using settings typical for high-complexity peptide analyses. Mass lock was set to “Best”. Full MS scans from 350 to 1400 m/z were acquired at 70 k resolution, with an AGC target of 3E6 and 100 ms maximum injection time. MS2 fragmentation was carried out on the top 10 precursors, excluding 1+ and >7+ charged precursors. The dynamic exclusion window was 30 s. Precursor isolation width was 1.4 m/z and NCE was 27. MS2 resolution was 17,500, with an AGC target of 5E5 and a maximum injection time of 50 ms. Protein identification was completed by MaxQuant using Swiss-Prot mouse proteome (version 2021_04) and default parameters. Label-free quantitation intensities were analyzed by the LFQ-Analyst pipeline to determine differentially abundant proteins based on *p*-values < 0.1 (Benjamini Hochberg adjusted *p*-value). Intensities were Z-score normalized and visualized in a heatmap.

### 2.6. Human Cohort Study

The project was approved by the Human Research Ethics Committees of QIMR Berghofer Medical Research Institute (QIMRB, P3675, approved 13 December 2020), New South Wales Health Pathology (NSWHP-RPAH 2020/ETH02630, approved 6 October 2020), and The Prince Charles Hospital (AM/2020/QPCH/63003, approved 24 March 2021). All participants provided written informed consent.

The cohort originated from three sites, and included (i) asymptomatic healthy volunteers (QIMRB and TPCH) that had not been in contact with COVID-19 cases for the past 14 days (COVID.NEG); (ii) COVID-19 positive (COVID.POS) hospitalized patients at TPCH, and (iii) COVID.POS individuals in hotel quarantine in New South Wales, Sydney, Australia. For cohorts (ii) and (iii), saliva sampling was performed within 14 days after the initial PCR diagnostic test.

Fasting prior to saliva collection was considered but dropped to align with the real-world screening scenario. Participants were requested to rinse their mouth with water and refrain from eating and drinking for 20 min prior to collecting 1.2 to 3 mL saliva as sublingual drool into a clean receptacle.

Samples from cohorts (i) and (ii) were stored on ice and processed within 30 min. After a brief vortex, an aliquot of raw saliva was transferred into a 1.5 mL Eppendorf tube and centrifuged for 10 min at 500× *g* at 4 °C to remove particulates. Clarified saliva was transferred into a cryotube containing ethanol to obtain 75% (*v*/*v*) ethanol, and incubated at room temperature for 30 min. Inactivated saliva samples were stored at −80 °C. For the TPCH COVID.POS samples, another tube was prepared to obtain 75% (*v*/*v*) Trizol for RT-PCR, using the protocol described above.

Samples from cohort (iii) were initially transported to the laboratory at room temperature. Aliquots of raw saliva were frozen at −80 °C, subsequently thawed on ice, inactivated with 75% (*v*/*v*) ethanol, and shipped on dry-ice to QIMR Berghofer for FTIR analysis. A nasal pharyngeal swab was collected on the same day as the saliva, and was analyzed by RT-PCR using the TaqPath COVID-19 Combo Kit (Thermo Fisher Scientific) according to manufacturer’s instruction.

For COVID.POS individuals, a subset tested PCR negative, and were thus classified as COVID.POS^FU.NEG^ (Appendix A).

### 2.7. ATR-FTIR Spectra Acquisition and Processing

Samples in 75% ethanol were thawed on ice and homogenized by high-speed vortexing. An aliquot of 2 µL was applied to the crystal of the ATR-FTIR instrument (Cary 630 FTIR, Agilent Technologies, Mulgrave, VIC, Australia) and allowed to air dry (~30 s) before spectral acquisition occurred over the wavenumber range 4000–650 cm^−1^. Background was collected without sample, i.e., ambient room air at 21 °C, between each measurement following cleaning of the crystal with 80% ethanol. Settings included 64 scans (Sample/Background) with a resolution of 8 cm^−1^. All spectra were baseline adjusted with baseline estimated using regions 2031–1865 cm^−1^ and 3971–3799 cm^−1^. Spectra were then normalized by adjusting the area under the curve (AUC) to 1.

### 2.8. Statistical Analysis

Euclidean distance was calculated for each pairwise comparison of normalized spectra to determine intra- and inter-sample variability. Each comparison was grouped into an “intra-sample” (spectra from same biological replicate, 1970 comparisons) or “inter-sample” (spectra from different biological replicate, 89,253 comparisons) category, and represented as a violin plot.

Clustering of the samples was explored using discriminant analysis to create a canonical plot to display clustering of clinical groups. The LogWorth statistic was applied to identify spectral regions that significantly deviate between two sample groups. The false discovery rate *p*-value cut-off for each comparison was chosen in a data-dependent manner accounting for the differences from the baseline.

A predictive model was developed using partial least squares discriminant analysis (PLS-DA) to predict clinical group based on the spectra. A seven-factor solution was chosen to account for at least 70% of the variation in the spectrum. A variable importance plot (VIP) was generated to indicate which areas of the spectrum contributed most to the predictive model. The fit of the model was evaluated using leave-one-out cross-validation (LOO-CV), generating a receiver operating characteristic (ROC) curve and a confusion matrix presenting the cross-validated sensitivity and specificity. The cutoff for predicting positive or negative results were chosen to maximize Youden’s Index (i.e., the sum of sensitivity and specificity).

## 3. Results

### 3.1. Characterisation of In Vitro SARS-CoV-2 Infection-Induced Secretome ATR-FTIR Spectra

As a first step, a standard Vero E6 cell in vitro infection model was used to investigate the secretory host response to SARS-CoV-2 infection. Two controls were used, a media control and an ultraviolet light (UV)-inactivated SARS-CoV-2, the latter of which could not replicate as UV destroys RNA. RT-qPCR of the SARS-CoV-2 RNA of the culture supernatant confirmed the lack of infectivity for both controls, while the active infection demonstrated an increased SARS-CoV-2 RNA load at 24 and 48 h (Figure 1a).

Interestingly, despite the 9-log increase in SARS-CoV-2 RNA at 24 h, the FTIR spectra of the supernatant showed minimal change at this time point, apart from increased absorbance at the amide I band (1700–1600 cm^−1^) in the active SARS-CoV-2 infected sample (Figure 1b). At 48 h, the FTIR profiles of UV-inactivated SARS-CoV-2 and active SARS-CoV-2 infected supernatants showed increased bands at 2970cm^−1^, 2924 cm^−1^, 2874 cm^−1^, 1590 cm^−1^, and 1415 cm^−1^, and decreased bands at 1373 cm^−1^, 1309 cm^−1^, 1042 cm^−1^, and 988 cm^−1^, compared to the media control (Figure 1b and Appendix A). However, at 48 h, the active SARS-CoV-2 infected secretome displayed separation from both controls in amide I/II bands (1700–1470 cm^−1^) and fingerprint (FP) region (1450–600 cm^−1^) (Figure 1b), as well as a right shift to a lower wavenumber from 1668 to 1595 cm^−1^ (Figure 1d and Appendix A).

FDR LogWorth analysis confirmed the significance for a number of these wavelengths from both controls, shown as the regions above the dotted line in Figure 1c (*p* < 0.001). To clarify the spectral changes for each condition over time, averaged spectra at the 48 h time point were subtracted from those at 24 h (Figure 1d and Appendix A). The greatest separations of spectra between active SARS-CoV-2 and UV-inactivated SARS-CoV-2 occurred at 2977 cm^−1^, 2920 cm^−1^, 1668–1665 cm^−1^, 1595 cm^−1^, 1418 cm^−1^, 1298 cm^−1^, 1122 cm^−1^, 1021 cm^−1^, and 854 cm^−1^ (Figure 1d and Appendix A). Active SARS-CoV-2 infection demonstrated separation from media control at 1600 cm^−1^, 1304 cm^−1^, 1124 cm^−1^, 1042 cm^−1^, and 1023 cm^−1^ (Figure 1c,d). These features notably included increased absorbance at 1124 cm^−1^, a region considered to reflect the symmetric stretching of phosphodiester linkages of RNA (ν_s_PO_2_^−^) [16,17].

### 3.2. ATR-FTIR Spectra of Oral Lavage from Respiratory SARS-CoV-2-Infected Mouse Model

Next, we used the transgenic ACE2 (K18-hACE2) mouse model, which develops lung infection and a respiratory disease resembling severe COVID-19 [18,19], and has been widely used to evaluate interventions against SARS-CoV-2 infection and disease [15,20,21,22,23,24]. Oral lavage was collected from anaesthetized mice prior to infection with SARS-CoV-2 or UV-inactivated SARS-CoV-2 (day 0), and then on days 2 and 4 post-inoculation (Figure 2a). The body masses of active SARS-CoV-2 infected mice started declining on day 3, reaching minus 10–15% on day 4 (Figure 2b). A comparison of the average oral lavage ATR-FTIR spectra showed more significant changes on day 4 compared to day 2 (Appendix A). On day 2, aliphatic, fatty acids, 1738 cm^−1^, and (ν_as_PO_2_^−^) bands showed increased absorbance with a mild drop in saccharides between groups (Appendix A). However, on day 4, saccharides dropped further, along with pronounced increases in the amide bands (Figure 2c,d and Appendix A). The subtraction of the spectra on day 4 from day 0 (baseline) allowed the visualization of respective time course changes for the UV-inactivated SARS-CoV-2 infection group (SARS-CoV-2^UV-I^) (Figure 2d, top) and the active SARS-CoV-2 infection group (SARS-CoV-2^POS^) (Figure 2d, bottom). Additional subtractive analysis was performed between these two groups to observe the unique changes attributed to active SARS-CoV-2 infection (Treatment, Figure 2d, middle). A broad, rising amide II peak at 1542 cm^−1^ was the most prominent feature resulting from SARS-CoV-2 infection. Although all mice displayed an increase in amide peaks between days 2 and 4, those with active infection presented a significant amide II peak and amide I shift (Figure 2c,d and Appendix A, *p* = 0.00001).

To further elucidate the pathophysiology, we examined the protein concentration and composition of the lavage. The elevated protein concentration in the SARS-CoV-2^POS^ lavage compared to the SARS-CoV-2^UV-I^ group indicated a strong secretory response to SARS-CoV-2 infection (Figure 2e). Proteomic analysis conducted on equal amount of protein from day 4 lavage samples. Due to limited protein, day 0 lavage samples could not be individually analyzed; hence, as a comparison, a pooled sample of day 0 lavage samples was prepared and analyzed. The proteins that were differentially abundant between SARS-CoV-2^UV-I^ and SARS-CoV-2^POS^ on day 4 were visualized in the heatmap in Figure 2f. This revealed upregulation of several kallikreins, and proteins involved in immune modulation, such as lectin galactoside-binding soluble 3 binding protein (LGALS3BP) and progranulin (Grn), while a number of proteins were comparatively downregulated, notably calmodulin 3 (Calm3). The oral lavage proteome for day 0 (pre-infection baseline) was broadly similar to day 4 of SARS-CoV-2^UV-I^ (Figure 2f).

### 3.3. ATR-FTIR Spectra of Human Saliva Distinguishes SARS-CoV-2 Infection Status

To investigate the application of ATR-FTIR for COVID-19 screening in human saliva samples, we collected saliva from 104 participants, comprising of 44 healthy controls (COVID.NEG) and 60 COVID-19 cases (COVID.POS) (Figure 3a, Appendix A). FTIR spectra were acquired in three to six technical replicates per biological sample, which were baseline corrected and normalized. Firstly, we assessed the technical variance of the method using pairwise Euclidean distancing (Appendix A). The variance within replicates of a participant was significantly lower compared to variance between participants (0.1925 ± 0.1941 vs. 0.6089 ± 0.544, *p* < 0.0001, Appendix A), indicating acceptable technical variability relative to the observed biological variability.

Visual inspection of the average spectra for COVID.POS and COVID.NEG groups showed promising differences in aliphatic, and amide I, II, III regions (Figure 3b), therefore, we next conducted discriminant analysis to visualize the separation by samples (Figure 3c). While COVID.NEG and COVID.POS groups separated on Canonical 1 (X-axis), we observed a separation within the COVID.POS groups along Canonical 2 (Y-axis) that correlate to the follow-up PCR results on the day of saliva sampling; termed COVID.POS^FU.POS^ and COVID.POS^FU.NEG^ (Figure 3c).

Next, we developed a partial least squares discriminant analysis (PLS-DA) model using the non-linear iterative partial least squares (NIPALS) algorithm to predict the three clinical groups. Seven factors explained 75.25% of the variation in the spectra. Based on the results from leave-one-out cross-validation (LOO-CV), the PLS-DA model correctly predicted 75% of COVID.NEG (specificity) and 93.48% of COVID.POS^FU.POS^ (sensitivity). While the sensitivity was in line with the previous cohort studies, the specificity was slightly lower in this cohort. Therefore, we investigated if recency of COVID-19 vaccination may contribute towards incorrect prediction by saliva FTIR. Of the 44 COVID.NEG participants, 29 participants had received one or two vaccine doses in the 8–120 days prior to saliva collection, with the four incorrectly predicted participants having vaccinations 22–67 days prior to saliva collection. As the time from vaccination to saliva collection was not recent, vaccination was unlikely to influence the saliva ATR-FTIR results.

While the PLS-DA model has high sensitivity for detecting COVID.POS^FU.POS^, its predictive performance for COVID.POS^FU.NEG^ on LOO-CV was poor. Of the 14 COVID.POS^FU.NEG^ participants, only two (14.3%) were predicted correctly, with the remaining 10 (71.4%) and two (14.3%) predicted as COVID.NEG or COVID.POS (Figure 3d). Accordingly, visual inspection of the average spectra revealed little separation between these subgroups (Figure 3e), but closer inspection (Appendix A) of the spectra revealed a separation between COVID.POS^FU.POS^ and COVID.POS^FU.NEG^ groups in the IR regions that we had previously reported to correlate with COVID-19 disease severity in plasma [12]. A larger sample size will be required to further investigate this result.

### 3.4. Delineation of Spectral Signature for COVID.POS^FU.POS^ Saliva

Two different statistical analyses were used to identify the COVID-19 saliva spectral signature for COVID-19 screening: LogWorth FDR analysis (Figure 4a,b) determines significantly different spectral regions between groups; variable importance plot analysis provides the relative contributions of predictive regions selected in the PLS-DA model (Figure 4c).

Figure 4a shows the LogWorth FDR analysis comparing COVID.NEG and COVID.POS^FU.POS^ saliva, which revealed significant differences in all amide bands, including increased absorbance in amide A and B (3500–3300 cm^−1^ and 3100 cm^−1^, respectively), a narrowing of amide I from a major right shift (1710–1650 cm^−1^) and minor left shift (1624–1596 cm^−1^), a pronounced increase and right shift of amide II (1570–1470 cm^−1^), and an increase of amide III (1320 cm^−1^). Significantly increased absorbances were also observed in aliphatic bands: 2956 cm^−1^ (ν_as_ CH_3_), 2870 cm^−1^ (ν_s_ CH_3_), 1464 cm^−1^ (δ_as_ CH_3_, asymmetric bending), 1420 cm^−1^ (δ CH_2_ and deformations), and 890 cm^−1^ (δ CH_2_). Bordering amide III, the two most significant combined points, 1252 cm^−1^ and 1228 cm^−1^, represent asymmetric phosphate stretching (ν_as_PO_2_^−^) among a variety of macromolecules, such as phospholipids, phosphorylated proteins, and RNA [25].

In addition, seven points of varying significance (*p* < 0.06–0.01) correlated with the significant peaks from the COVID.POS^FU.POS^ vs COVID.NEG comparison: right shift of amide I (1688/1658 cm^−1^); decreased aliphatic/RNA (1430 cm^−1^); decreased δCH_3_^−^ bending (1373 cm^−1^); decreased ν_s_PO_2_^−^ RNA (1124 cm^−1^), ν_s_PO_2_^−^, symmetric, and C-O ν ribose (1071 cm^−1^); decreased νC_4_-OH^−^ glucose (1016 cm^−1^). The right shifting amide I peak in COVID.POS^FU.POS^, compared to both COVID.NEG and COVID.POS^FU.NEG^, is in agreement with residual misfolded amyloid protein fibrils and elevated IgA in COVID-19 patient saliva (Appendix A) [9,26,27].

Unsurprisingly, few statistically significant differences were observed in the LogWorth FDR analysis comparing COVID.POS^FU.POS^ and COVID.POS^FU.NEG^ saliva (Figure 4b). The comparison of the LogWorth FDR analysis with the VIP analysis of the PLS-DA model (Figure 4c) revealed that most of the predictive peaks overlapped with the significant peaks from the COVID.POS^FU.POS^ vs COVID.NEG analysis (Figure 4a).

### 3.5. Comparison of COVID-19 Spectral Signature across Diverse Models

Finally, we sought to establish the most characteristic COVID^POS^ spectral signature across multiple models and studies through a meta-analysis of significant spectral bands from all three study arms, as well as the three recent publications. Despite the differing nature of the models and methodologies across studies, several consistent spectral changes due to SARS-CoV-2 infection were identified (Table 1). Most strikingly, a change in the structure of proteins was indicated by an amide II increase in all studies, indicative of β-sheet structures. In all human cohorts (but not in in vitro or mouse models), amide III, aliphatic, phosphodiester asymmetric stretching (ν_as_PO_2_^−^), and saccharide bands were also increased. In contrast, saccharide bands were decreased in in vitro and mouse models, with VIP values 1.2–1.43 over the range 1067–1006 cm^−1^ (Figure 3b,e and Appendix A). This range of wavenumbers was previously recognized for having significance in predicting severe COVID-19 outcomes when evaluating blood plasma, including an elevated AUC at 1592–1588 cm^−1^ [12]. Taken together, these results suggest a change in the physiological response between end-point/severe COVID-19 (in vitro and mouse models, and severity study) and COVID-19 patients who are tolerating the disease well.

## 4. Discussion

This study provides several missing links needed to support the translation of ATR-FTIR spectra for COVID-19 saliva screening, from a simple, consistent sample processing method to controlled infection experiments that delineate background signals. An important, but previously unreported, result from the current human cohort study is the separation of saliva FTIR spectra based on the follow-up SARS-CoV-2 status at the time of sampling. This novel finding has implications for the application of a saliva screening test, but require increased study size. Nevertheless, the consistency of the COVID.POS FTIR signature across the in vitro, mouse, and human arms of this study, and three existing independent reports [8,9,10], provides confidence that this robust technology is ready for clinical development and deployment.

Compared to existing and developing COVID-19 tests [4,5], the proposed FTIR test is unique in providing broad biochemical information, rather than relying on a single specific measurement of SARS-CoV-2 RNA or protein. Our results show that the saliva FTIR spectra captures the pathophysiological response to SARS-CoV-2 infection, which supports the use of saliva as a non-invasive, self-collectable bio-sample reflective of the physiological response [34,35,36], and may have both diagnostic and prognostic utility. Our cohort study, along with the three 2021 COVID saliva/pharyngeal swab FTIR studies [8,9,10], independently developed predictive models with high specificity and sensitivity for detecting positive COVID-19 cases from healthy individuals. We have already reported a proof-of-concept study for the use of FTIR in COVID-19 disease severity prediction using plasma [12]. While the ‘severity’ peaks were observed in this study, a clinical study with follow-up outcome data will be required to establish the utility for COVID-19 prognosis from saliva FTIR spectra.

The COVID-19 saliva FTIR signature shares many biochemical features with amyloid deposits (aliphatic, amide, and phosphodiester ν_as_) and lipofuscin [26,37]. Enhanced amyloid formation in SARS-CoV-2 has been a recent area of research focus [38,39,40]. The consistent COVID-19-associated amide absorbance shifting from an α-helix (~1652 cm^−1^) to a β-sheet (~1636 cm^−1^) composition is likely related to SARS-CoV-2 induction of protein aggregates through its spike protein [41,42,43]. Strikingly, our proteomics data revealed extensive upregulation of kallikrein proteins in the oral lavage of SARS-CoV-2^POS^ mice. Savitt et al. recently reported direct interaction and activation of the kallikrein/kinin system (KKS) by recombinant SARS-CoV-2 proteins S, M, N, and E [44]. High molecular weight kininogen (HK) and plasma prekallikrein (PK) bring about the sequelae of bradykinin, and complexing of HK/PK with blood coagulation factor XII (FXII), initiate the intrinsic clotting cascade with the aid of misfolded proteins and polyphosphate [45]. Polyphosphate may serve as a natural defense blocking the receptor binding domain for SARS-CoV-2 [46]. Whether derived from platelets or commensal bacteria, the extensive utilization of polyphosphate in the KKS/FXII pathogenesis of SARS-CoV-2 offers another substantial explanation for the phosphate and amide III profiles among these studies [47,48,49]. Another upregulated protein contributing to the amide I/II bands, LGALS3BP, is potentially upregulated as a compensatory defensive mechanism to the prolonged innate immune response by day 4 [50]. Further investigations should be carried out to establish protein disaggregated in human saliva samples and the involvement of KKS-FXII-polyphosphate.

While the cell culture and mouse model data were generally consistent with our human cohort data, the reduced saccharide band in cell and mouse SARS-CoV-2 treated samples contrasted with the increases observed in human COVID.POS samples. This discrepancy was most probably due to different stages in the evolution of infection, as the mouse and cell experiments represent models of severe illness, while the human cohorts consisted of individuals where SARS-CoV-2 infection was generally well-tolerated, and many subjects were entering the recovery phase. Studies have provided mechanistic evidence for the metabolic dysregulation in COVID-19, notably through insulin resistance involving adiponectin/leptin and proinflammatory alterations [51,52], which fits with our observations, along with the decreased food intake observed during the acute phase of human disease, i.e., as evidenced by the infected mice in our studies [52].

A novel finding from our patient cohort is the separation of the COVID.POS patients into a sub-group with low/undetectable viral load on the day of saliva collection, based on their saliva FTIR spectra. Saliva FTIR spectra of this COVID.POS^FU.NEG^ group displayed reduced saccharide (ribose) bands (1038 cm^−1^ and 1074 cm^−1^) compared with COVID.POS^FU.POS^. These bands coincided with the ATR-FTIR bands for extracted SARS-CoV-2 RNA [53], in agreement with the PCR results; however, this difference in saccharide absorbance may also indicate recovery from the previously described, hyperglycemic state. This group also showed reduced signal in the finger print region, proposed by Martinez-Cuazitl et al. [9] to represent immunoglobulins IgG, IgM, and IgA. As these COVID.POS^FU.NEG^ patients are likely to have continued immunoglobulin expression/secretion [54], our results suggest that the amide I/II and fingerprint regions more likely correlate with the clearance of protein aggregates (β-sheet) and aliphatic amino acids, as shown by the significant decreases at 1688 cm^−1^ and 1373 cm^−1^, respectively (Figure 4b).

While the predictive model using saliva ATR-FTIR spectra showed high sensitivity in predicting COVID.NEG and COVID.POS^FU.POS^ cases, it was unable to differentiate COVID.POS^FU.NEG^ cases accurately. Nevertheless, when thinking how one may utilize this technology as a possible point-of-care screening test, the high sensitivity makes it able to rule out infected individuals who are likely to transmit SARS-CoV-2. This utility is extremely important from an epidemiological and social perspective as the world aims to return to “normal”, with infection control measures in place for gatherings, work places, and schools [55].

In contrast to a previous saliva FTIR study that required fasting for >8 h prior to saliva collection [9], we took a pragmatic approach of only 20–30 min abstinence from food prior to testing. Our results support this time interval between sample collection and testing, making a point-of-care rapid testing application more feasible. It is unlikely that this time interval can be shortened further as saliva is likely to be contaminated with food particles, thus interfering with FTIR signals. We did notice, however, excessive precipitation while mixing saliva with ethanol, secondary to the initial high postprandial cephalic secretion. Adding low-speed centrifugation of raw saliva prior to inactivation with ethanol circumvented this problem. Our simple inactivation procedure with ethanol removes any possible biosafety concerns. All these features make the future development of point-of-care application feasible. However, saliva collection and processing methods would require additional refinement (e.g., use of a capillary action sampling cartridge).

The strengths of this translational medicine study, aimed to support the translation of a COVID-19 FTIR saliva test, include the use of well-controlled infection models to shed light on the pathophysiological basis of the test, and the integration of data from diverse model and independent human cohort studies to demonstrate the robustness of the technology. The limitations to be addressed in future studies include the small cohort sample size and lack of an independent cohort for validation of the predictive model.

## 5. Conclusions

In conclusion, ATR-FTIR technology with saliva self-collection provides a simple, rapid, and biosafe sample processing procedure, which has high potential as a non-invasive, low-resource method for COVID-19 screening. The simplicity of the method means that only basic skills are required to conduct the test, which would satisfy the global need for rapid COVID-19 screening at diverse locations, such as airports and public venues. Further evaluation may also establish the utility for COVID-19 prognosis. As the method requires only generic laboratory equipment, ethanol, an ATR-FTIR instrument with implemented predictive algorithm, and a power source, it offers promise as a global tool in the management of the COVID-19 pandemic.

## Figures and Tables

**Figure 1 biomedicines-10-00351-f001:**
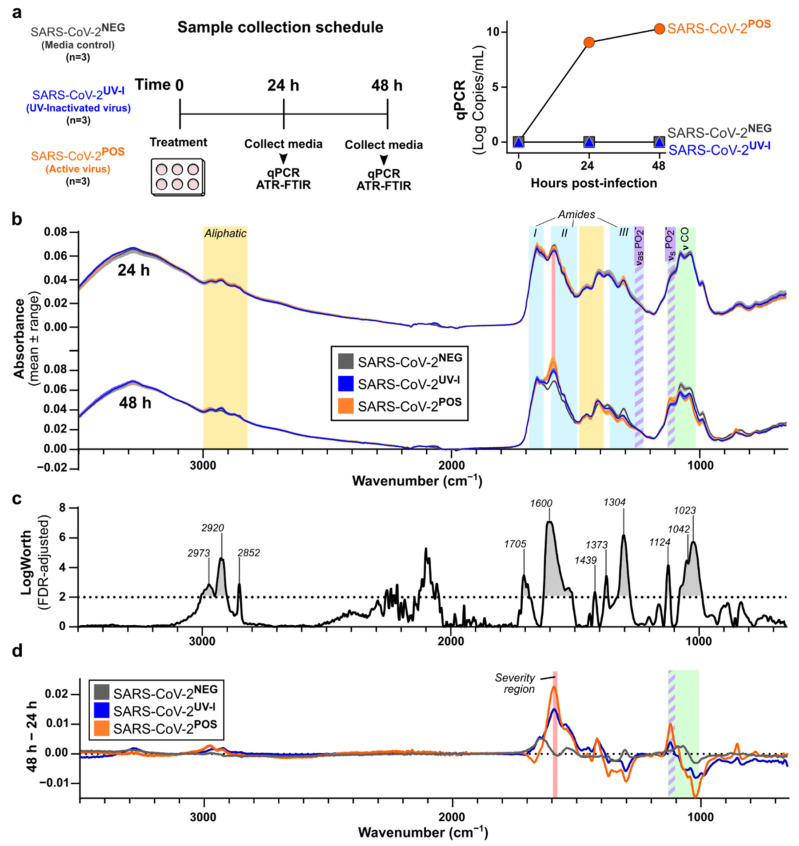
ATR-FTIR spectral changes of culture supernatants of in vitro SARS-CoV-2 infection model. (**a**) Vero E6 cells (6 × 10^5^ total cells) were treated with media alone (SARS-CoV-2^NEG^), UV-inactivated (SARS-CoV-2 ^UV-I^), or SARS-CoV-2 (SARS-CoV-2^POS^) for 2 h, after which cells were washed with PBS and media replaced. Aliquots of conditioned media were collected at 24 h and 48 h post-infection for qPCR and ATR-FTIR. Verification of viral load was accomplished via RT-qPCR (*p* = 0.0035). (**b**) Overlapping spectra of technical replicates for 24 h and 48 h time points. Colored bands indicate chemical components of interest: aliphatic (yellow), amide I/II/III (cyan), severity [12] (red), saccharide (green), phosphodiester asymmetric stretching (ν_as_PO_2_^−^) and symmetric stretching (ν_s_PO_2_^−^) (purple stripes). (**c**) Significant features of SARS-CoV-2 infection compared to the two controls at 48 h, using FDR LogWorth analysis; dotted line represents FDR LogWorth 2 (*p* < 0.01). (**d**) Subtraction of supernatant spectra for each treatment from 24 h to 48 h.

**Figure 2 biomedicines-10-00351-f002:**
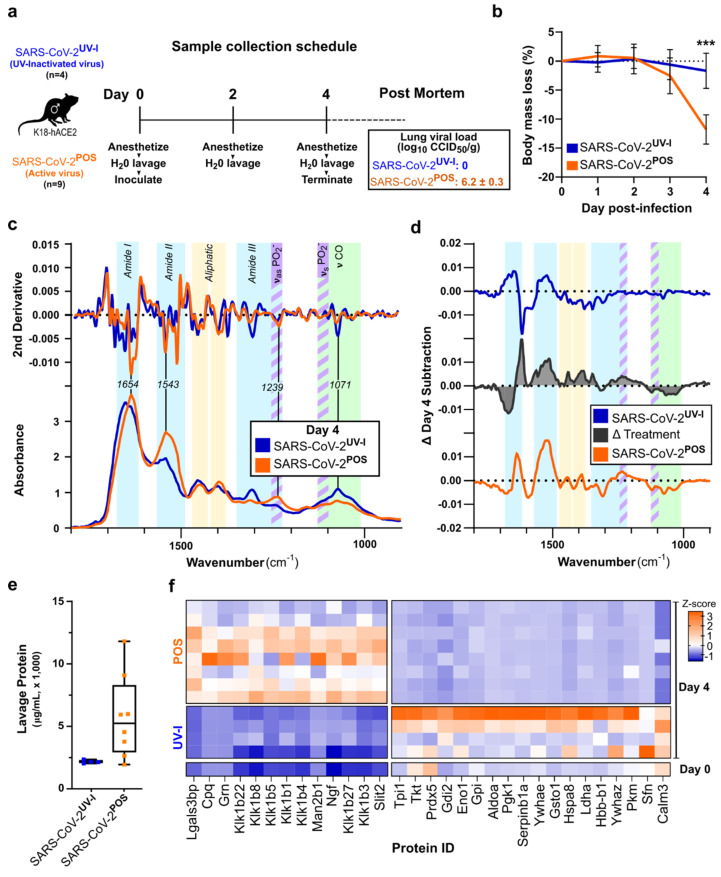
ATR-FTIR spectra and proteomic changes of oral lavage of SARS-CoV-2 mouse model. (**a**) Male K18-hACE2 mice were inoculated with intrapulmonary UV-inactivated (*n* = 5) or active SARS-CoV-2 (*n* = 9), and oral lavage was sampled on days 2 and 4. Viral load of mouse lung tissue was assessed via cell culture infectious dose 50% assay (CCID_50_) post-mortem, showing no active virus in the inactivated virus group. (**b**) Body weight measurements were recorded daily. Error bars show standard error. *** on day 4, *p* = 0.0004. (**c**) Day 4 oral lavage ATR-FTIR spectra of the amide I/II and fingerprint regions with respective 2nd derivative (above). Colored bands indicate chemical components of interest: amide (protein) bands I, II, III (cyan), PO_2_^−^ asymmetric (ν_as_) and symmetric (ν_s_) stretching (purple stripes), saccharides (green), with identification of key peaks by wavenumber. (**d**) Subtraction of day 4 spectra from day 0, showing a time course alteration for SARS-CoV-2^UV-I^ (blue) and SARS-CoV-2^POS^ (orange), as well as the difference between the groups, Δ Treatment (black). Complete spectra (4000–600 cm^−1^) as well as day 2 data are available in Appendix A. (**e**) Protein concentration of day 4 oral lavage plotted per group. (**f**) Proteomics was conducted on equal amounts of day 4 oral lavage, and on a pooled sample of day 0 oral lavage (3 samples) for comparison. Heatmap shows Z-scores of differential proteins (*p* < 0.1 adjusted) between SARS-CoV-2^UV-I^ and SARS-CoV-2^POS^ groups.

**Figure 3 biomedicines-10-00351-f003:**
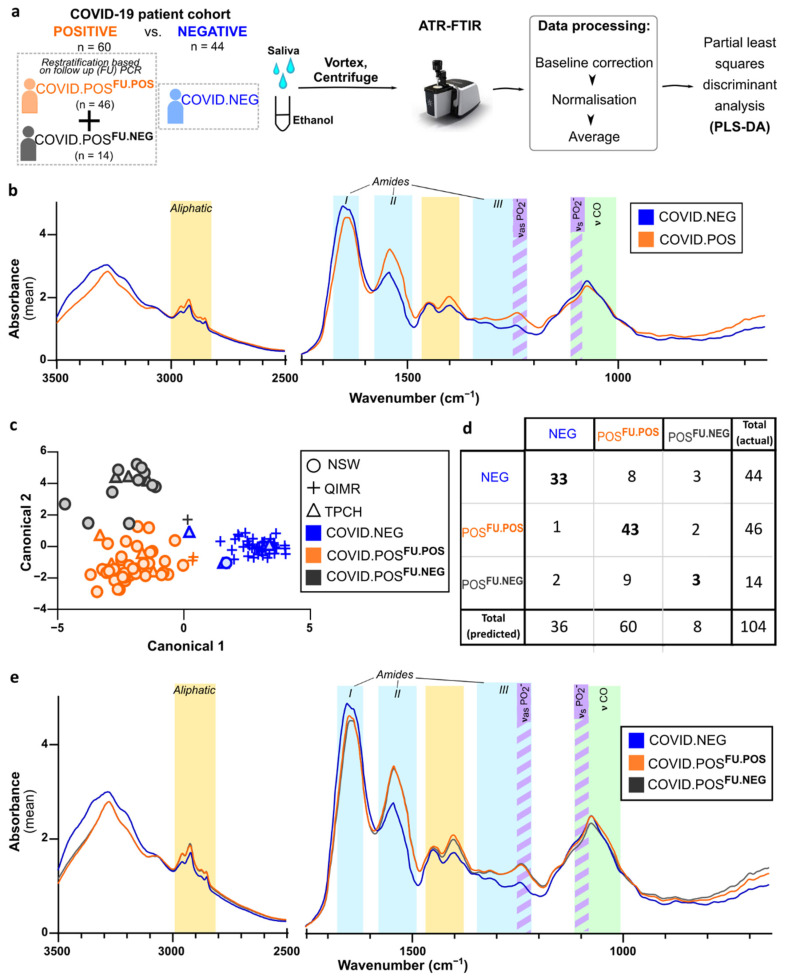
ATR-FTIR spectral data of human cohort. (**a**) Workflow. Sublingual saliva samples were collected from human subjects with known COVID.POS and COVID.NEG status. Follow-up SARS-CoV-2 PCR was conducted for the COVID.POS group on the saliva or swab collected on day of saliva collection (COVID.POS^FU.POS^ or COVID.POS^FU.NEG^). Clarified saliva adjusted to a final concentration of 75% ethanol was used for ATR-FTIR on an Agilent Cary 630 FTIR, with samples dried (~30 s) on the crystal. Data for each technical replicate were baseline corrected, then normalized to an AUC of 1. (**b**) Average spectra (3500–650 cm^−1^) of COVID.NEG and COVID.POS groups. Colored bands indicate chemical components of interest: aliphatic (yellow), amide I/II/III (cyan), saccharide (green), phosphodiester (purple stripes). (**c**) Canonical plot with symbols indicating the location of sampling; NSW, New South Wales Health Pathology; TPCH, The Prince Charles Hospital; QIMRB, QIMR Berghofer Medical Research Institute. (**d**) Contingency table for leave-one-out cross-validation of the partial least squares discriminant analysis (PLS-DA) model. Columns represent actual designation, while rows represent predicted categorization. Bold numbers indicate the correct prediction. (**e**) Average spectra (3500–650 cm^−1^) for each of the three clinical groups. Labelling as for panel (**b**).

**Figure 4 biomedicines-10-00351-f004:**
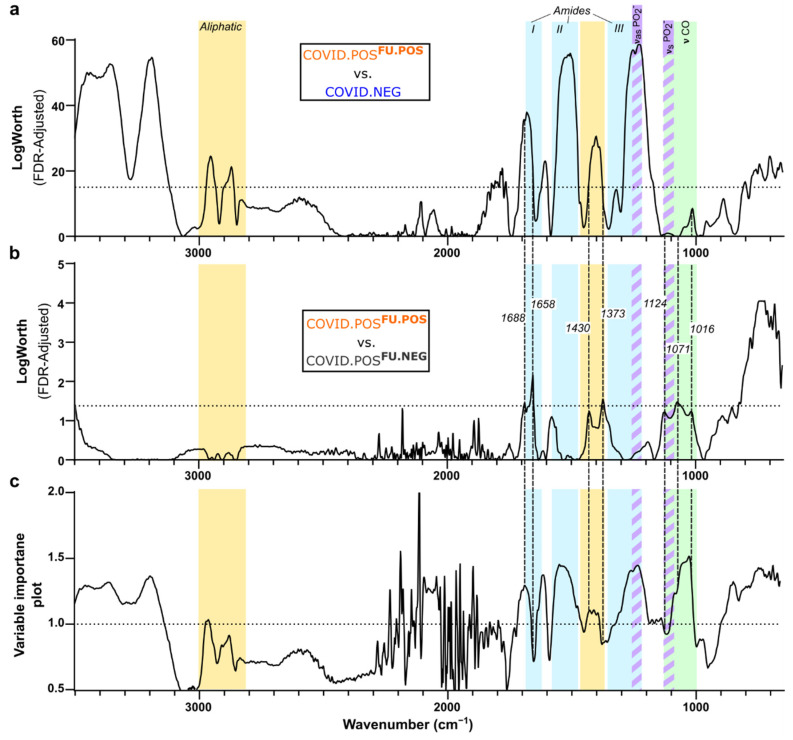
Delineating FTIR spectral signature for COVID-19 saliva screening. Saliva FTIR spectra from the cohort analysis in Figure 3 was subjected to comparative FDR LogWorth analysis for (**a**) COVID.POS^FU.POS^ and COVID.NEG saliva samples, and (**b**) COVID.POS^FU.POS^ and COVID.POS^FU.NEG^ samples. Dotted lines mark the cut-off above noise at LogWorth of 15 (*p* = 1 × 10^−15^) and 1.3 (*p* = 0.05), respectively. (**c**) Variable importance plot for the final PLS-DA model shown in Figure 3d. Dotted line indicates VIP of 1.0. Colored bands indicate chemical components of interest: aliphatic (yellow), amide I/II/III (cyan), saccharide (green), phosphodiester (purple stripes).

**Table 1 biomedicines-10-00351-t001:** FTIR spectral features for saliva/secretion in COVID^POS^ cohorts/models.

Band Designation	In Vitro ^a^	Mouse ^a^	Human ^a^	Chemical Components [26,28,29,30,31,32,33]	Barauna [8]	Wood [10]	Martinez- Cuazitl ^c^ [9]
Amide A		**3246**	3358	N-H, O-H stretching		X	
			3518–3280 ^b^				
Amide B		**3067**	3190	Amide II overtone, aromatic amino acids		X	
			3248–3110 ^b^				
Aliphatic	**2973**	2931–2880	**2954**	-CH_3_/-CH_2_; C-H symmetric (ν_s_) & asymmetric (ν_as_) stretching		X	
		2858	**2870**				
		2837	2968–2944 ^b^				
Fatty Acids	1705	1702	1722–1704	-COOH, C = O ν; and ketones		X	O
			1714–1690 ^b^				
* Amide I	1690		1680 ^b^	Protein β-sheets; C = O guanine			
		**1638**		Protein β-sheets			
	**1600**		1625–1594	Protein aggregates; amyloid fibrils			X
			1632–1585 ^b^				
Amide II	**1524**	**1578–31**	**1572–1470**	N-H; primarily β-sheet		X	X
Aliphatic Fingerprint		1468	**1464**	-CH_2_ δ, bending vibrations		X	
	**1439**	1431	**1416**	-CH_2_ δ, symmetric stretching band of carboxyl group, CH_2_ ω, wagging; RNA	X	X	X
			**1420** ^b^				
			**1402**	C–H deformation; CH_2_ ω; C–N stretching; In-plane C2′OH in RNA		X	X
			**1400** ^b^				
	1373	1370	**1375**	-CH_3_ δ, C-H ν; methyl bending/stretching		X	X
			**1388–1376** ^b^				
Amide III	1302	1302	**1319**	Amino acid side-chains; terminal oxygen (PO_3_^-^)		X	X
		1378–1354	**1340–1285**	-CH_2_ ω; -CH_3_ δ, amyloid contribution		X	X
		1335–1280	**1330–1177** ^b^				
			**1250**	PO_2_ ν_as_; C-N ν		X	X
			**1250** ^b^				
			**1243–1218**	PO_2_ ν_as_; amyloid fibrils		X	X
			**1226** ^b^				
RNA	**1124**		1129 ^b^	PO_2_ ν_s_, phosphodiester stretching		X	
Saccharide		1094		-C-O-C, ether linkages; -O-Ca^2+^ c	X		
	1072	1064	1077 ^b^	PO_2_^−^ ν_s_, symmetric and C-O ν	X	X	X
	1050			-C-O ν, C-OH group; C-C ν (sugars)		X	X
	1023		**1034–1003**	C-O ν; P-O ν; C-OH δ			X
			**1095–997** ^b^				
	1008	1012	**1012**	C_4_-OH, Glucose		X	X
	988	**988–974**		PO_2_^−^ ν_s_; -C-O-, ribose		X	
			**940**	P-O ν, phosphorylation; -C-C- ν		X	
		**887–866**	**889**	P-O-P ν_as_; -C-C- ν; aromatics		O	X
		830	**836** ^b^	P-O-C ν; = C-H δ; aromatics		O	X

^a^ Significant by FDR LogWorth analysis. Bold type indicates where SARS-CoV-2 spectra was higher; ^b^ Also significant by variable importance plot analysis; **^c^** Long fasting collection; analysis of truncated spectra.; * Some observed significance in amide I region was due to a shift of spectral peaks; X, separation of spectra from control; O, noticeable change in spectra, value not reported.

## Data Availability

Proteomics data are available via ProteomeXchange with identifier PXD030012. Raw FTIR spectra and associated clinical data are available at DOI: 10.5281/zenodo.5703689, https://zenodo.org/record/5703689#.YZLxWGBBxaQ (uploaded on 15 November 2021).

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
