# Peer review of "Pathophysiological Response to SARS-CoV-2 Infection Detected by Infrared Spectroscopy Enables Rapid and Robust Saliva Screening for COVID-19"

_biomedicines, 2022, doi:10.3390/biomedicines10020351_

Round 1
Reviewer 1 Report
Manuscript by Kazmer et al describes use of FTIR spectroscopy for COVID-19 saliva screening. The authors used infected cell culture supernatant, and mouse oral lavage and detected covid infection induced FTIR signals over negative control background. Further, proteomic analysis of mouse oral lavage revealed upregulation of several immunomodulatory proteins and down regulation of proteins like Calmodulin -3. Authors extended these stufies with human cohort and obtained the FTIR spectral data of saliva collected from Covid negative and positive patients; the spectra showed differences between the aliphatic and amide regions between the two. Predictive model developed, however failed to correctly predict Covid patients negative after follow up. Based on their data and those published the authors also identified several spectral changes characteristic to Covid-positive samples, consistent across studies, suggesting that use of FTIR spectra is a promising tool for covid saliva screening
This is a well performed study, the methodology and results are clearly presented and discussed.
Author Response
Thank you for the positive feedback.
Reviewer 2 Report
The topic of the paper is interesting and very up-to-date. The literature basis enough and the research methodology is well described.
The paper itself has some flaws worth improving:
- Describe the goal of the paper.
- Describe the research question.
- Pleas add more information to literature review part – it should have 3 pages and be based on more good international journals papers.
- Please describe the links between the research gap and the goal of the paper and research question. Write why the paper is important. What is the main contribution of the paper to the field?
- The discussion part in the paper is not enough linked with international literature – in this part should be analysis of relations between results and literature.
- Add limitation of the paper.
Round 2
Reviewer 2 Report
Authors have implemented my remarks.
Author Response
Thank you again for your time reviewing our paper.